# Exposure to a Low-Oxygen Environment Causes Implantation Failure and Transcriptomic Shifts in Mouse Uteruses and Ovaries

**DOI:** 10.3390/biomedicines12051016

**Published:** 2024-05-05

**Authors:** Asmaa Y. Ammar, Fatma M. Minisy, Hossam H. Shawki, Mohamed Mansour, Shabaan A. Hemeda, Abeer F. El Nahas, Ahmed H. Sherif, Hisashi Oishi

**Affiliations:** 1Biotechnology Department, Animal Health Research Institute (AHRI), Agriculture Research Center (ARC), Kafrelsheikh 12619, Egypt; asmaa_ammar86@yahoo.com; 2Genetics Laboratory, Department of Animal Husbandry and Animal Wealth Development, Faculty of Veterinary Medicine, Alexandria University, Alexandria 21544, Egypt; shabaan.hemeda57@yahoo.com (S.A.H.); abeer.elnahas@alexu.edu.eg (A.F.E.N.); 3Department of Comparative and Experimental Medicine, Nagoya City University Graduate School of Medical Sciences, Nagoya 467-8601, Japan; fatma.m.minisy@gmail.com; 4National Gene Bank of Egypt, Giza 12916, Egypt; 5Fish Disease Department, Animal Health Research Institute (AHRI), Agriculture Research Center (ARC), Kafrelsheikh 12619, Egypt

**Keywords:** hypoxia, hypoxia-inducible factor (HIF), implantation, corpus luteum, progesterone

## Abstract

Hypoxia is a condition in which tissues of the body do not receive sufficient amounts of oxygen supply. Numerous studies have elucidated the intricate roles of hypoxia and its involvement in both physiological and pathological conditions. This study aimed to clarify the impact of a forced low-oxygen environment in early pregnancy by exposing mice to low-oxygen conditions for 24–72 h after fertilization. The treatment resulted in the complete failure of blastocyst implantation, accompanied by vascular hyperpermeability in the uterus. A transcriptome analysis of the uterus revealed remarkable alterations in gene expression between control normoxic- and hypoxic-treatment groups. These alterations were characterized by the differentially expressed genes categorized into the immune responses and iron coordination. Furthermore, exposure to a low-oxygen environment caused apoptosis in the corpus luteum within the ovary and a reduction in progesterone secretion. Consequently, diminished plasma progesterone levels were considered to contribute to implantation failure in combination with the activation of the hypoxic pathway in the uterus. Additionally, previous studies have demonstrated the impact of hypoxic reactions on blastocyst development and the pre-implantation process in the endometrium. Our findings suggest that the corpus luteum exhibits elevated susceptibility to hypoxia, thereby elucidating a critical aspect of its physiological response.

## 1. Introduction

Oxygen is essential for the production of energy in physiological processes in mammals [1]. Oxygen functions as a final electron acceptor in the process of the oxidative phosphorylation of glucose, where ATP is produced in mitochondria [2]. A lack of oxygen in cells can lead to a condition known as hypoxia, which can disrupt normal cellular functions and ultimately result in cell damage or death [3]. The cellular response to hypoxia is a complex process that involves both physiological and pathological adaptations, such as angiogenesis, glycolysis, and apoptosis [3]. For example, hypoxic conditions are essential for maintaining hematopoietic stem cells, bone formation, and homeostasis [4,5]. Pathological changes including solid tumors, inflammatory process, and tissue fibrosis are associated with the hypoxic response [6].

Hypoxic responses are mainly mediated by α/β heterodimeric transcription factors known as hypoxia-inducible factors (HIFs) [7]. The two principal HIF-α isoforms, HIF-1α and HIF-2α, are widely expressed and form distinct heterodimeric complexes (HIF-1 and HIF-2) with the dominant HIF-β isoform, HIF-1β. HIF-1α and HIF-2α show 48% similarity at the amino acid level, and HIF-1 and HIF-2 induce the common sets of gene expression [8]. These two transcription factors also induce unique sets of gene expression due to the differences in the expression of cells and post-translational modifications [8]. HIF-1 is widely expressed and induces the expression of glycolytic genes, some pro-angiogenic genes, and genes involved in pH regulation [9]. HIF-2 is expressed in specific types of cells including endothelial cells, cardiomyocytes, kidneys, and hepatocytes, and facilitates the expression of matrix metalloproteinases and erythropoietin genes [9]. In addition, HIF-1 is activated immediately after the onset of hypoxia, whereas HIF-2 is activated more slowly and sustained [8]. Thus, HIF-1 and HIF-2 have a different role in their function.

In humans, embryogenesis takes place in a hypoxic environment (2–3% or 15–20 mmHg of O_2_ levels) for the first 10 weeks of pregnancy due to low blood circulation [10]. The hypoxic microenvironment is considered to maintain the phenotypes of trophoblast stem cells and undifferentiated cells by means of metabolic and cell-proliferation controls [11]. Additionally, a cluster of extravillous trophoblasts invades both the maternal decidua and the lumen of the spiral arteries, initially plugging the arteries to prevent high-velocity blood flow to the placental surface. The plug in the spiral arteries creates a physiological hypoxic environment that is beneficial to trophoblast differentiation and placental development.

An increasing number of studies have demonstrated the significance of the oxygen environment in inflammation and tissue remodeling [12]. In the female reproductive system, the presence of a hypoxic placenta is considered to be associated with preeclampsia [13]. Additionally, animal studies have shown that the induction of chronic hypoxia leads to hypertension and proteinuria similar to preeclampsia, accompanied by an activation of a hypoxic response in the placenta and a reduction in trophoblast invasion. In preeclampsia, the hypoxic response causes various effects that affect placentation, endothelial function, inflammation, oxidative stress, and angiogenesis [14]. Although many studies have shown an impact of hypoxia in the second to third trimesters related to preeclampsia, few studies have attempted to study it just after fertilization. In particular, little research has been conducted concerning the pre-implantation period. A previous report, in which uterus-specific *Hif-2α* knockout mice showed infertility with impaired embryo invasion, suggests the importance of examining the hypoxic effect during the pre-implantation period [15].

In this study, we examined the effect of a hypoxic environment on early pregnancy in vivo. The pregnant mice treated in the low-oxygen condition exhibited a defective implantation of blastocysts, a disturbed pre-implantation process of the uterus, and apoptosis of the corpus luteum in the ovary. In addition, the transcriptome analysis of uteruses and ovaries revealed remarkable alterations in gene expression between the control normoxic- and hypoxic-treatment groups. Our findings suggest that the corpus luteum exhibits an elevated susceptibility to hypoxia, followed by a diminished plasma progesterone levels and destructive blastocyst implantation, thereby elucidating a critical aspect of its physiological response.

## 2. Materials and Methods

### 2.1. Animal Care and Ethics Statement

The experiment was approved and conducted in a human manner according to institutional animal care and use committee (IACUC) guideline of Nagoya City University, Japan (No. 21-002). C57BL/6NCrSlc mice (10–14 weeks of age) were purchased from Japan SLC (Shizuoka, Japan), and housed under specific pathogen-free conditions. Mice were housed in a temperature-controlled room at 22 ± 2 °C on a 12 h light/dark cycle, and fed standard chow and water ad libitum.

### 2.2. Experimental Design

The mice were mated and the presence of a vaginal plug at the next morning was designated as gestational day (GD) 0.5. To induce a hypoxic environment, the mice were sequestered in a chamber at GD0.5, subjected to evacuation using a vacuum pump (DA-60S, ULVAC, Kanagawa, Japan), and the atmospheric pressure was consistently maintained at 700 hPa, which corresponds to approximately 70% of the O_2_ amount at the average sea level (Appendix A). This condition was maintained for 24–72 h from GD0.5.

### 2.3. Blood and Tissue Collection

The mice were anesthetized with an inhalation of isoflurane gas. Blood was collected from the retro-orbital sinus in EDTA-containing tube. Ovary and uterine horns were excised and flushed with PBS for the collection of blastocysts, and fixed in 4% for histological analyses.

### 2.4. Histology and Immunofluorescence Study

Fixed tissues underwent a process of dehydration and subsequent paraffin embedding. Sections, 5 µm in thickness, were stained with hematoxylin-eosin. For the Ki-67 immunofluorescence study, sections were incubated for 30 min in a permeabilization buffer (0.3% TritonX-100 in PBS), and then incubated in an antigen-retrieval buffer (10 mM, sodium citrate, pH 6.0). The sections were blocked with a blocking solution (10% rabbit serum in PBS) and incubated with Ki-67 antibody (ab15580; Abcam, Cambridge, UK), followed by the detection with anti-rabbit IgG conjugated with FITC (Abcam). Finally, the sections were mounted with a DAPI-containing mounting medium (Thermo Fisher Scientific, Waltham, MA, USA), and the images were acquired using a confocal laser scanning microscope (FV3000; Olympus, Tokyo, Japan).

### 2.5. Plasma Progesterone Level Measurements

Collected blood was centrifuged at 1500× *g* for 15 minutes at 4 °C, and the supernatant was stored at −20 °C. The plasma progesterone levels were measured by using progesterone ELISA kit (ADI-900-011; Enzo Life Science, Farmingdale, NY, USA), according to the manufacturer’s instructions. The absorbance of the sample and standards was measured at 405 nm using a SpectraMax 340PC microplate reader (Molecular Devices, Sunnyvale, CA, USA).

### 2.6. TUNEL Assay

Apoptotic DNA fragmentation was examined using the MEBSTAIN Apoptosis TUNEL kit Direct (MBL, Tokyo, Japan) according to the manufacture’s protocol.

### 2.7. Quantitative RT-PCR Analysis

Total RNA was extracted from uterine and ovarian tissues using ISOGEN II (Nippongene, Tokyo, Japan). In total, 1 μg of extracted RNA was converted to cDNA using a QuantiTect Reverse Transcription Kit (Qiagen, Hilden, Germany). Real-time quantitative PCR was performed using PowerUp SYBR Green Master Mix (Thermo Fisher Scientific, Waltham, MA, USA) in a QuantStudio 12K Flex real-time PCR system (Thermo Fisher Scientific). The expression level of the genes was normalized to *Hprt* using the ΔΔCt method. Primer sequences are shown in Appendix A.

### 2.8. RNA-seq Analysis

Total RNAs were extracted from 3 mice of each group and pooled. RNA-sequencing libraries were constructed using the NEBNext Poly(A) mRNA Magnetic Isolation Module (New England Biolabs, Ipswich, MA, USA), and MGI Easy RNA Directional Library Prep Set (MGI Tech, Shenzhen, China) following the manufacturer’s instructions. The indexed libraries were combined and sequenced on a DNBSEQ-T7RS platform (MGI Tech) with a 2 × 150 base paired-end configuration. Subsequently, FASTQ files were aligned to mouse reference genome (mm10), and quantified using the Galaxy platform. Differentially expressed genes (DEGs) were identified based on a log2-fold change (FC) threshold of ≥2.0. The Gene Ontology analysis of the identified DEGs was performed using the Enricher web tool at https://maayanlab.cloud/Enrichr/ (accessed on 25 March 2024) 

### 2.9. Statistical Analysis

The results are presented as means ± standard errors of the mean (SEM). Comparisons between the two groups were carried out using an unpaired Student’s *t*-test. A *p* value that was <0.05 was considered significant difference.

## 3. Results

### 3.1. Low-Oxygen Environment Increased the Expressions of Vegfa

To examine whether housing in a low-oxygen chamber induced a hypoxic response in the uterus, *Vegfa* expressions, which is the directly transactivated by HIFs, were examined. Mice were placed in the chamber for 24 h from GD0.5, and the uteruses were collected at GD1.5 for quantitative RT-PCR analysis (Appendix A). The treatment significantly induced the expression of *Vegfa* by 2.7 ± 0.5-fold, indicating that the treatment induced a local hypoxic response in vivo. *Hif-2α* expression was not transcriptionally changed by the treatment (Appendix A).

### 3.2. Low-Oxygen Environment Interfered with the Tissue Remodeling of Uterus and Blastocyst Implantation

To examine the blastocyst implantation, a Chicago blue dye solution was intravenously injected, and we counted the number of spotted bands as an indication of implantation sites. Although control mice showed 8.3 ± 0.7 blue stains in the uterus which corresponded to the implantation sites, there were no spotted stains in the hypoxic group of mice (Figure 1A,B). We also noticed that a whole blue-colored uterus was observed only in the hypoxic group, indicating the presence of extensive vascular permeability. Uterine histology also exhibited the tissue edema in the stroma of the hypoxic group of mice (Figure 1C). We also collected non-implanted blastocysts by flushing uteruses with PBS at GD4.5. No blastocysts were collected from the control group of mice because of the successful implantation, whereas a number of blastocysts were recovered from the hypoxic group of mice, suggesting implantation failure (Figure 1D). However, the recovered blastocysts manifested abnormal development. In addition, the hypoxic group of mice showed a decrease in the expression of genes involved in implantation including leukemia inhibitory factor (*Lif*), *Hand2*, and *Hoxa11* in uteruses at GD3.5 (Figure 1E).

The RNA-seq analysis of uteruses demonstrated remarkable alterations in gene expression between the control normoxic and hypoxic groups (Figure 2A). We identified 491 up-regulated and 412 down-regulated transcripts with a log2-fold change of ≥2.0, indicating significant differential expression between the control and treatment groups (Appendix A). To validate the RNA-seq data, we conducted a qRT-PCR analysis on a subset of randomly selected genes (Appendix A), which demonstrated consistent results. Further analysis of Gene Ontology for the up-regulated genes revealed that among the top significantly enriched biological pathways were certain elements involved in the defense response to fungus, iron-coordination entity transport, and cytolysis by the host of symbiont cells (Figure 2B), while down-regulated genes were enriched for the positive regulation of leukocyte tethering or rolling, tyrosine phosphorylation of the STAT protein, and the regulation of calcium-ion-dependent exocytosis (Figure 2C). Although GO analyses did not reveal the direct composition related to impaired implantation, the number of differences might contribute the phenotype in an integrated manner.

### 3.3. Low Progesterone Level Caused a Defective Receptivity during Pre-Implantation Period

Successful blastocyst implantation requires progesterone (P_4_) bursts from the ovarian corpus luteum after ovulation and the subsequent remodeling of the uterus endometrium. Then, we examined plasma P_4_ levels of the control and hypoxic groups of mice. The results showed that the low-oxygen treatment failed to reach comparative levels of P_4_ in plasma to the control (Figure 3A). The genes involved in the ovarian steroidogenesis pathway, namely *Hsd3b1* and *Cyp11a1*, showed a significant reduction in their expression levels (Figure 3B). Next, to find the possible causes of the decrease in P_4_, the degree of apoptosis in the corpus luteum was examined by using a TUNEL assay. TUNEL-positive cells were observed only in the hypoxic group of mice (Figure 3C). In addition, the rise in *Bax*-to-*Bcl-2* ratio, signifying a pro-apoptotic environment, supports the occurrence of apoptosis within the corpus luteum in the ovaries of mice exposed to hypoxia (Figure 3D).

RNA-seq analysis demonstrated remarkable alterations in gene expression between the control normoxic and hypoxic groups of ovaries, in which 433 and 421 transcripts were up- and down-regulated by a ≥2.0 log2-fold change, respectively (Figure 4A, Appendix A). The RNA-seq data accuracy was validated through conducting a qRT-PCR analysis on a random gene subset (Appendix A). Further Gene Ontology analysis revealed that up-regulated genes were significantly enriched in the regulation of cytosolic calcium-ion concentration and the positive regulation of leukocyte degranulation (Figure 4B), while down-regulated genes were enriched in histidine catabolic/metabolic process and intermediate filament organization (Figure 4C). As matched to the results in Figure 3B, the genes involved in the steroid hormone biosynthetic process were included in the list of down-regulated genes in the hypoxic group of mice.

Meanwhile, in the group of mice exposed to a low-oxygen environment, luminal epithelial cells in the uterus continued to exhibit proliferative activity, while decidual stromal cells showed a reduction in their proliferative activity (Figure 5A). The reduction of *Ihh*, which is the downstream target of the P_4_ signal, was consistent with the previous results in the uteruses of the hypoxic group of mice. In contrast, *Ltf* and *Lcn2*, which are induced by estrogen signal, were up-regulated (Figure 5B). These findings are indicating that an insufficient amount of P_4_ could not prime the uterine-receptivity process in hypoxic mice. Altogether, the data suggest that the corpus luteum was sensitive to the hypoxic environment and underwent apoptosis, ultimately impacting the implantation process.

## 4. Discussion

In this study, we examined the effect of a low-oxygen environment corresponding to a high altitude on early pregnant mice. The mice showed that the treatment in low-oxygen conditions induced the hypoxia-responsive pathway in the uterus, resulting in destructive implantation failure. We also found that the ovary, especially the corpus luteum, was sensitive to the hypoxia, leading to apoptosis and impaired progesterone secretion. The RNA-seq data support the idea that a low progesterone release from the corpus luteum is responsible for the insufficient preparation of the receptive endometrium necessary for successful embryo implantation.

A previous study shows that the expression levels of *Hif-1α*, which is primarily expressed in luminal epithelium, does not change throughout pregnancy, whereas *Hif-2α* is strongly induced in subluminal stromal cells after embryo attachment [16]. Uterine-specific *Hif-1α* and *Hif-2α* knockout mice show subfertility and infertility, respectively [15]. In the peri-implantation mouse uterus, HIFs are expressed in response to ovarian steroid hormones; P_4_ stimulates *Hif-1α* expression in the luminal epithelial cells, and estrogen stimulates *Hif-2α* in uterine stroma [16]. Our results, in which the reduction of P_4_ secretion from the corpus luteum impaired uterine receptivity and normal blastocyst development treated by the low-oxygen condition, were consistent with the previous findings. Taken together, a systemic low-oxygen environment mainly affected the hormonal production in the corpus luteum.

A number of animal studies have shown that a hypoxic condition affects ovary development and function. In non-pregnant mice, hypobaric hypoxia induces a prolonged estrous cycle, atretic follicles, and the reduction of estrogen in plasma [17]. In addition, hypoxic treatment promotes luteal cell death and a reduction in P_4_ in the bovine corpus luteum without the alteration of BCL2 and BAX expression [18] Interestingly, female fish exposed to hypoxia affected the F2 offsprings, which are not exposed to hypoxia. This study demonstrated that the F2 generation of the hypoxia-treated F0 generation developed reproductive impairments including atretic follicle and retarded oocyte development, due to the alteration of the DNA methylation pattern [19,20]. Thus, a hypoxic environment affects uterus function and embryonic development beyond animal species, although the mechanisms might be different among species. In this regard, we believe that our results provide important insights into the effects of hypoxic environments on the maintenance of early pregnancy.

The RNA-seq data in this study show an increase in the number of genes involved in immune responses in hypoxic uteruses. It is well known that tumor hypoxia regulates a variety of immunological responses in addition to angiogenesis. Although we did not examine details of the type of inflammatory cell filtration and cytokines, the local immune response probably inhibited the pre-implantation process and subsequent implantation. The inability to address possible treatments or clues for hypoxia in humans is also a limitation of this study.

It has been reported that hypoxia exhibits an association with various gynecological diseases. Notably, preeclampsia, in its pathogenesis, manifests a significant linkage with placental hypoxia [13]. In this context, it is worth nothing that the placental tissues of afflicted patients demonstrate a heightened expression of HIF-1α, thereby affording a mechanism for the augmented transcription of genes pivotal to the disease’s pathophysiology [21]. Furthermore, the etiology of endometriosis is also intricately intertwined with the hypoxic pathway [22]. Multiple alterations originating from the hypoxic pathway actively contribute to its pathogenesis, encompassing the dysregulation of cyclooxygenase (COX)-2 expression, the amplification of estrogen signaling, and the induction of neo-vascularization in endometriotic tissues [23]. Additionally, in women afflicted with heavy menstrual bleeding (HMB), a decrease in endometrial *Hif-1α* was demonstrated in a recent study [24]. These associations between hypoxia and human diseases are also supported by animal models. For instance, a rat model of preeclampsia has been established through deliberate reduced uterine perfusion pressure (RUPF), accomplished through a surgical intervention that restricts blood flow through the abdominal aorta and ovarian arteries [25]. Furthermore, a number of studies have convincingly demonstrated the influence of hypoxia on rodent and primate models of endometriosis [26]. There are no human diseases directly corresponding to the animal phenotypes in this study; however, a low-oxygen environment might recapitulate acute high-altitude, low-oxygen conditions. The atmospheric pressure in this study was maintained at 700 hPa, which equivalates to an altitude of 3000 m. In total, 140 million people are considered to live their lives higher than 2500 m all over the world [27]. Interestingly, it is reported that living at a high altitude is a significant risk factor of preeclampsia [28]. Therefore, our experimental system could be improved for a novel preeclampsia model by means of controlling oxygen levels and duration.

In this study, we examined the effect of hypoxia on early pregnancy in vivo. Our findings suggest that the corpus luteum exhibits an elevated susceptibility to hypoxia, followed by diminished plasma progesterone levels and destructive blastocyst implantation, thereby elucidating a critical aspect of its physiological response.

## Figures and Tables

**Figure 1 biomedicines-12-01016-f001:**
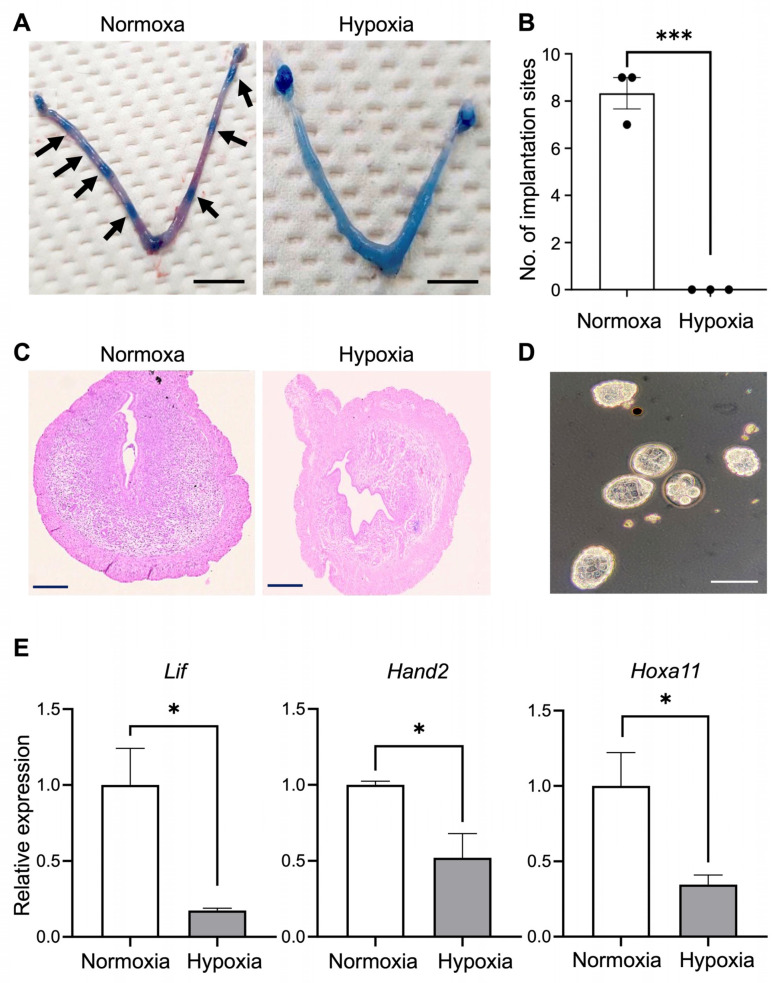
Hypoxic environment caused implantation failure in a pregnant mouse. (**A**) Implantation sites (arrows) were grossly visible in normoxic but not hypoxic uterine horns at GD4.5 after an intravenous injection of Chicago blue dye. Scale bars: 1.0 cm. (**B**) The number of implantation sites of uteruses in normoxic (n = 3) and hypoxic environments (n = 4) of mice. (**C**) Representative uterine images of hematoxylin and eosin (H&E), showing decidualization of stromal cells around an implanting embryo in normoxic but edema in hypoxic uterine sections at GD4.5. Scale bars: 500 µm. (**D**) Blastocysts recovered from uterine horns of hypoxic mice at GD4.5. Scale bars: 100 µm (**E**) mRNA levels of *Lif*, *Hand2*, and *Hoxa11* in the normoxic and hypoxic mice. * *p* < 0.05. *** *p* < 0.001.

**Figure 2 biomedicines-12-01016-f002:**
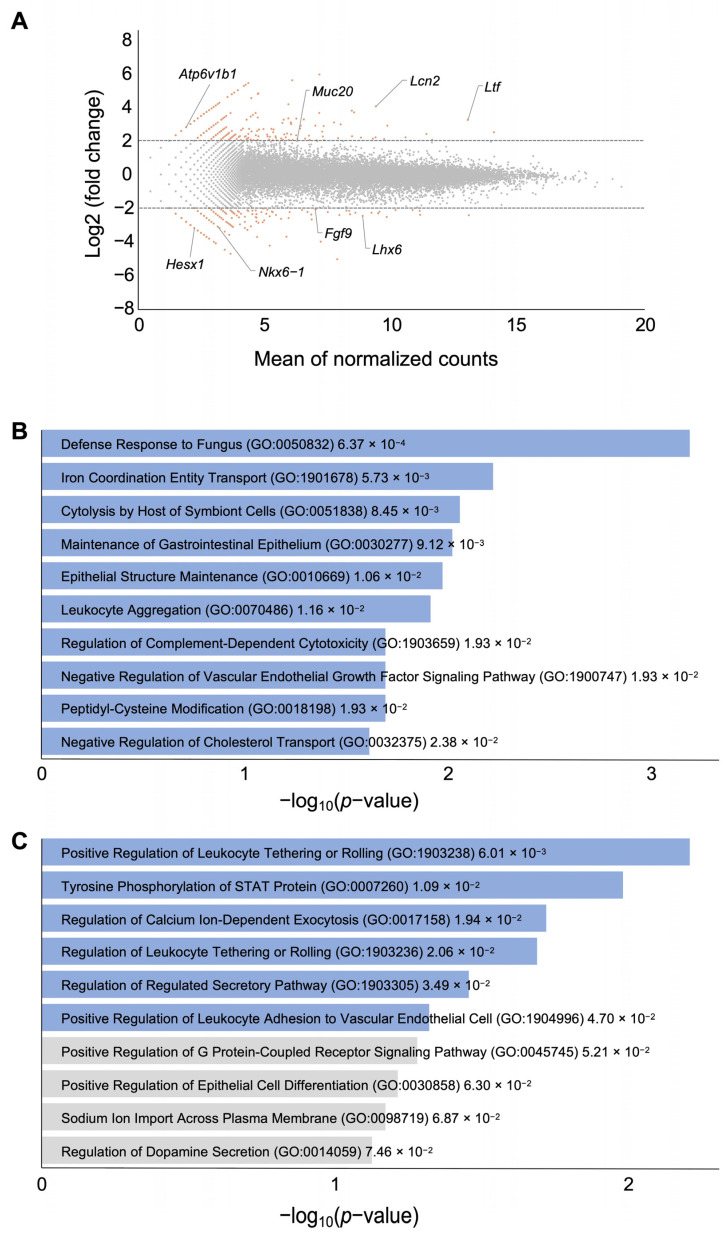
Gene expression profiles of control and hypoxic uteruses at 3.5 GD. (**A**) MA-plot of RNA-seq data, where the log2-fold change is plotted against the mean of normalized counts for genes. Dashed lines indicate up-regulated and down-regulated genes with a threshold of ≥2.0-fold. Colored dots correspond to terms with ≥2.0-fold. (**B**,**C**) The top ten Gene Ontology (GO) enrichment based on the *p*-value in terms of biological function for the up-regulated (**B**) and down-regulated (**C**) genes of hypoxic uteruses revealed by Enrichr analysis using DEGs from RNA-seq. Actual *p*-values are shown next to each term. Colored bars correspond to terms with significant *p*-values (*p* < 0.05).

**Figure 3 biomedicines-12-01016-f003:**
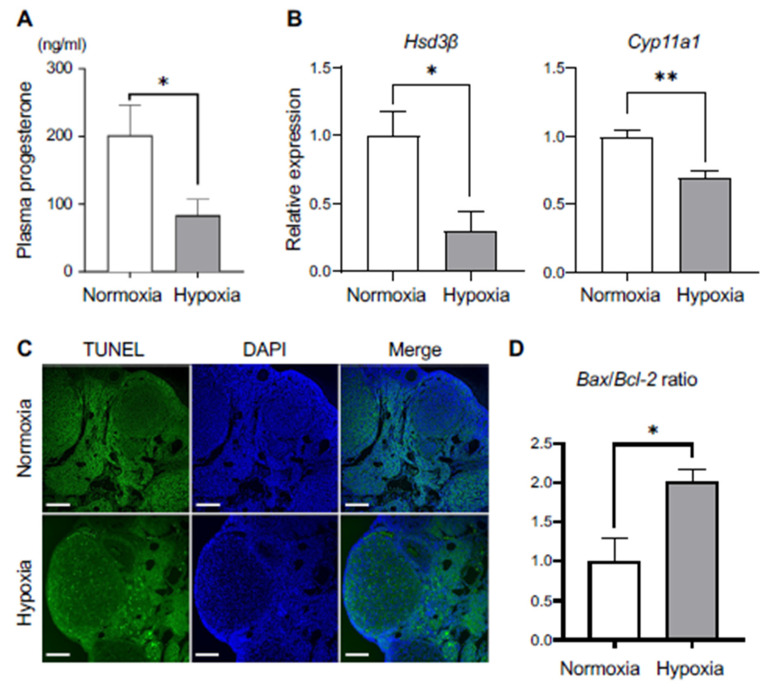
Low plasma progesterone disrupted uterine receptivity in the mice under hypoxic environment. (**A**) Plasma progesterone (P_4_) levels in normoxic and hypoxic environments (n = 5 each group) at GD3.5. * *p* < 0.05. (**B**) mRNA levels of *Hsd3β* and *Cyp11a1*. * *p* < 0.05. ** *p* < 0.01 (n = 3; control group, n = 4; hypoxic group). (**C**) TUNEL staining of the ovaries in the normoxic and hypoxic environments. Scale bars: 100 µm. (**D**) *Bax*/*Bcl-2* mRNA ratio of the ovaries in the normoxic and hypoxic environments at GD3.5 (n = 4 each). * *p* < 0.05.

**Figure 4 biomedicines-12-01016-f004:**
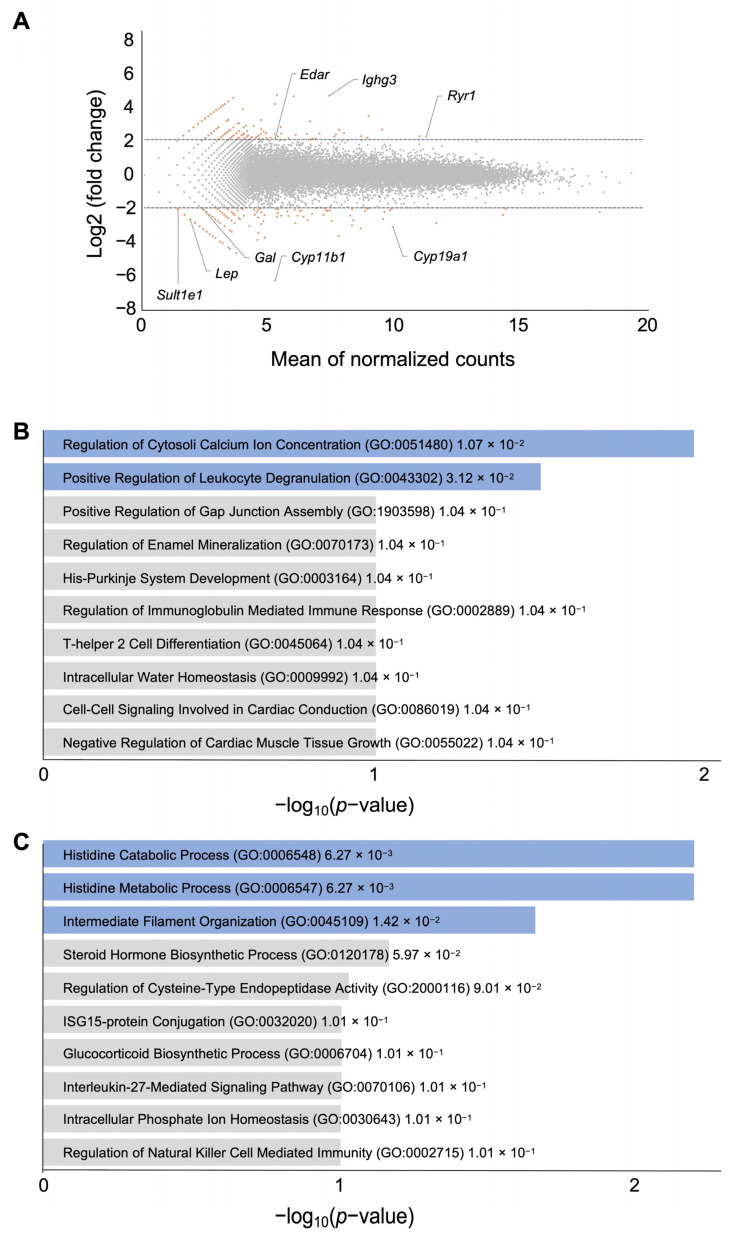
Gene expression profiles of ovaries in control and hypoxic environments at GD3.5. (**A**) MA-plot of RNA-seq data, where the log2-fold change is plotted against the mean of the normalized counts for genes. Dashed lines indicate up-regulated and down-regulated genes with a threshold of ≥2.0-fold. Colored dots correspond to terms with ≥2.0-fold. (**B**,**C**) The top ten Gene Ontology (GO) enrichments based on the *p*-value in terms of biological function for the up-regulated (**B**) and down-regulated (**C**) genes of hypoxic ovaries. Actual *p*-values are shown next to each term. Colored bars correspond to terms with significant *p*-values (*p* < 0.05).

**Figure 5 biomedicines-12-01016-f005:**
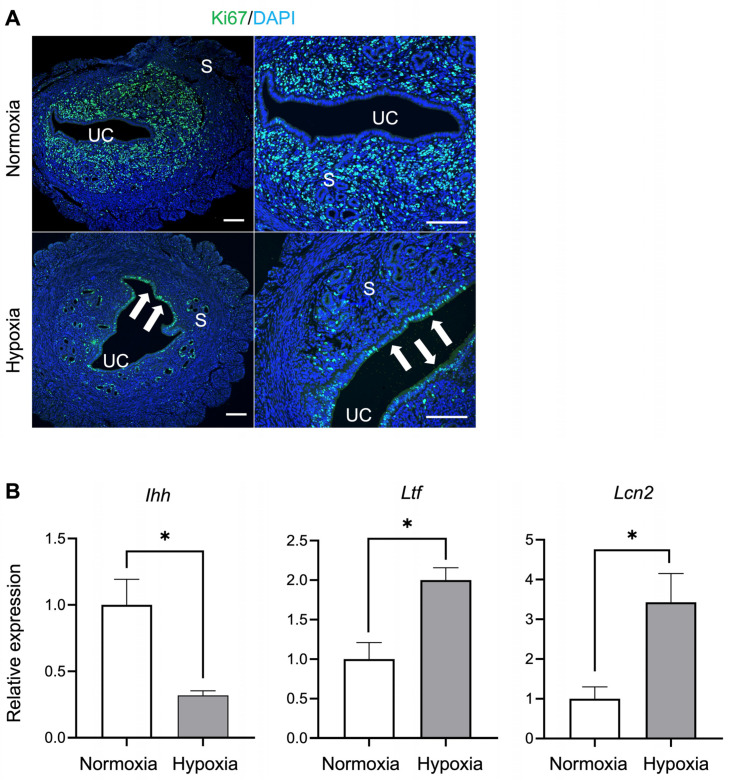
Reduction of plasma progesterone prevented receptivity in hypoxic uterus at GD3.5. (**A**) Representative immunofluorescence staining of Ki67 and DAPI at GD3.5. Arrows indicates Ki67-positive cells in luminal epithelial cells. Scale bars: 100 µm. UC: uterine cavity. S: stroma cells. (**B**) mRNA levels of *Ihh*, *Ltf*, and *Lcn2*. * *p* < 0.05. (n = 3; control group, n = 4; hypoxic group).

## Data Availability

The data presented in this study are available on request from the corresponding author.

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
