# Peer review of "Exposure to a Low-Oxygen Environment Causes Implantation Failure and Transcriptomic Shifts in Mouse Uteruses and Ovaries"

_biomedicines, 2024, doi:10.3390/biomedicines12051016_

Round 1

Reviewer 1 Report

Comments and Suggestions for Authors

The manuscript currently meets the criteria to be considered for publication.

You just have to check that when you use an abbreviation for the first time it is defined.

Author Response

We appreciate your suggestion. We have checked all abbreviations and added complete expressions for the first time.

Reviewer 2 Report

Comments and Suggestions for Authors

Content suggestions:

1.         What are the suggestions of the Authors for the practice ?

2.         Did the Authors perform a study in which the hypoxia would be overcomed by iron supplementation or LMWH administration in the case of thromboembolism as the cause of hypoxia to make the recommendations for humans ?

The article is written on a high scientific level with the use of sophisticated methods reaching the molecular level. I sincerely appreciate the effort of the Authors to provide the new information in this field of study.

Author Response

We appreciate valuable comments.

1, 2. We completely agree with the reviewer’s concern; however, we could not address the possible treatment and human clinical situations. Therefore the points have been discussed as a limitation of the study in the discussion section as follows (line 300).

"The inability to address possible treatments or clues for hypoxia in human is also a limitation of this study."

Reviewer 3 Report

Comments and Suggestions for Authors

Exposure to a Low-Oxygen Environment Causes Implantation Failure and Transcriptomic Shifts in Uterus and Ovary

Reviewer statement:

Oxygen is essential for the production of energy and a lack of oxygen, also known as  hypoxia disrupt normal cellular functions and leads to cell damage. As stated by the authors, in humans, embryogenesis takes place in a hypoxic environment for the first 10 weeks of pregnancy due to low blood circulation. The hypoxic state is beneficial for the pregnancy development , including trophoblast differentiation and placental development. Hypoxia in the second to third trimesters is related to the preeclampsia. There is a lack of studies who examine the effect of hypoxia after fertilization and in the pre-implantation period. The authors conducted a study to the effect of hypoxic environment on early pregnancy, which is interesting and relevant.

Title:

The title chosen reflects the study being reported, reflecting the study being performed.

1.        The authors should considered adding , an animal study  to the title, as this makes clear to the reader what type of study has been conducted.

Overall:

The article was pleasant to read with overall good English grammar.

Abstract

The abstract section is well written. The abstract should be adapted based on the provided comments and suggestions.

Introduction  

The introduction section is well written, stating the background of performing study. It was pleasant to read, explaining well the background and the reason for conducting this study.

No comments on this section.

Materials and methods

This section is well written and good to understand. Nevertheless, some important points need clarification and/ or explanation.

2.       The authors report in line 105-108: “ To induce a hypoxic environment, the mice at GD0.5 were sequestered in a chamber, subjected to evacuation using a vacuum pump (DA-60S, ULVAC, Kanagawa, Japan), and the atmospheric pressure was consistently maintained at 700 hPa. As a reader, it was not clear if the describe condition should be considered as hypoxic. As only the atmospheric pressure is describe. Was the oxygen level still at 21%? Please elucidate on the condition to help the reader.

3.       Moreover, is the describe condition considered a condition to qualify as hypoxic? As this is not standard knowledge and familiar to most readers, the authors should help the reader in understanding the condition. This is essential for the interpretation of the presented results.

Results

The result section is well written and was pleasant to read. Nevertheless, an important point need clarification and/ or explanation.

4.       The authors report in line 157-158: “ To examine whether housing in low-oxygen chamber induced hypoxic response in uterus, Vegfa expressions, which is the directly transactivated by HIFs, were examined..  As a reader the statement low-oxygen chamber should also be explained, see remarks in point 2 and 3.

Discussion

This discussion section is well written and was pleasant to read. In this section the authors address crucial and important factors and points, including limitation. The authors elucidate on important factors for the interpretation of the presented result, and correlate the result to humans, which is excellent.

No comments on this section.

Figures and tables:

No comments.

Author Response

We appreciate your helpful comments. We have responded to all the comments raised by a professional reviewer individually.

I. We have changed the title to "Exposure to a Low-Oxygen Environment Causes Implantation Failure and Transcriptomic Shifts in Mouse Uterus and Ovary."

2, 3.  To clarify the experimental condition, we have added following explanation in the end 'which is approximately 70% of the O2 amount at the sea level.' In addition, the following explanation has been added at line 265-265. "In this study, we examined the effect of a low-oxygen environment corresponding to a high altitude on early pregnant mice."

4. The photo of chamber and explanation have been added in the supplementary figure.